# A Comprehensive Survey of Prospective Structure-Based Virtual Screening for Early Drug Discovery in the Past Fifteen Years

**DOI:** 10.3390/ijms232415961

**Published:** 2022-12-15

**Authors:** Hui Zhu, Yulin Zhang, Wei Li, Niu Huang

**Affiliations:** 1Tsinghua Institute of Multidisciplinary Biomedical Research, Tsinghua University, Beijing 102206, China; 2National Institute of Biological Sciences, 7 Science Park Road, Zhongguancun Life Science Park, Beijing 102206, China; 3RPXDs (Suzhou) Co., Ltd., Suzhou 215028, China

**Keywords:** structure-based virtual screening, molecular docking, structural novelty

## Abstract

Structure-based virtual screening (SBVS), also known as molecular docking, has been increasingly applied to discover small-molecule ligands based on the protein structures in the early stage of drug discovery. In this review, we comprehensively surveyed the prospective applications of molecular docking judged by solid experimental validations in the literature over the past fifteen years. Herein, we systematically analyzed the novelty of the targets and the docking hits, practical protocols of docking screening, and the following experimental validations. Among the 419 case studies we reviewed, most virtual screenings were carried out on widely studied targets, and only 22% were on less-explored new targets. Regarding docking software, GLIDE is the most popular one used in molecular docking, while the DOCK 3 series showed a strong capacity for large-scale virtual screening. Besides, the majority of identified hits are promising in structural novelty and one-quarter of the hits showed better potency than 1 μM, indicating that the primary advantage of SBVS is to discover new chemotypes rather than highly potent compounds. Furthermore, in most studies, only in vitro bioassays were carried out to validate the docking hits, which might limit the further characterization and development of the identified active compounds. Finally, several successful stories of SBVS with extensive experimental validations have been highlighted, which provide unique insights into future SBVS drug discovery campaigns.

## 1. Introduction

Structure-based virtual screening (SBVS), also known as molecular docking, has been increasingly applied to discover novel ligands for a target of interest in the early stage of drug discovery efforts [1,2]. In particular, the molecular docking approach is designed to identify small molecules complementary to the target’s ligand-binding pocket with a known 3D structure from a large chemical library. The critical issues of molecular docking include generating suitable ligand binding poses and scoring and ranking compounds based on the estimated binding affinities. The highly ranked hits are then carefully inspected, evaluated, and subjected to experimental evaluation [3,4].

Generally, the success of a practical virtual screening campaign can be judged by the true positive rate of experimentally tested compounds, the structural novelty of the identified active compounds, the binding affinity, and drug-likeness properties. Recent review articles have highlighted the developments in docking methodology [5,6,7,8], and relatively narrow areas of docking applications [9,10,11,12]. In this review, we did not attempt to provide an exhaustive summary of all relevant studies on molecule docking. Instead, we focused on systematically surveying and analyzing 419 prospective SBVS case studies reported in the past fifteen years and critically evaluating the quality of the docking screening by objective criteria, including but not limited to the novelty of the protein targets and docking hits, the applied docking screening protocols, and the types of experimental validations. Our comprehensive survey and analysis will help the community obtain a bigger picture of SBVS in early drug discovery and distinguish between high-quality and low-quality studies.

The most significant advantage of molecular docking is its ability to effectively and efficiently identify novel chemotypes from a large chemical library against a target of interest. Usually, novel chemotypes unrelated to previously known ligands for a given target might confer new biological consequences. Furthermore, the meaningful differentiation of chemotypes could provide unique properties to answer additional questions that known ligands couldn’t address in biological studies. To better understand this advantage of docking screening, we specifically analyzed the docking performance against new targets and the structural novelty of the docking hits. Finally, four representative SBVS case studies are described in detail. We anticipate that this review paper will offer valuable guidance for understanding, evaluating, and applying molecular docking screening in future practical applications.

## 2. Literature Survey

We used the keyword “virtual screening” or “in silico screening” to retrieve the records in the past fifteen years from the ChEMBL database [13,14] and obtained 1338 documents. We also used the same keywords to survey the Research Collaboratory for Structural Bioinformatics Protein Data Bank (RCSB PDB) website [15,16] and obtained 693 records. In addition, we augmented 65 articles curated from different resources that could not be retrieved from ChEMBL and PDB databases using the keywords mentioned above. Furthermore, we carefully inspected these articles and excluded those that did not meet our selection criteria, resulting in 419 case studies (Appendix A). Firstly, the primary library screening had to be target-based docking screening. Therefore, we excluded any work that relied on pharmacophore-based approaches, machine-learning algorithms, ligand-based similarity searches, quantitative structure-activity relationship (QSAR) models, and a pre-defined chemical scaffold (for example, piperidine derivatives). Secondly, the work has to be a prospective study with solid experimental validation (i.e., well-defined potency endpoints, including IC_50_, EC_50_, Ki, and Kd). Publications reporting hits with single-point measures like “percentages inhibition” were excluded. Thirdly, we excluded work focusing on lead optimization, retrospectives studies, and review articles. Finally, the articles must clearly describe the virtual screening protocol.

We summarized the information manually extracted from the literature into three parts (Appendix A):Basic information: the article title, the publication year, the google scholar citation, and the institution of the corresponding author.Docking details: the target name, the target classification, the docking PDB structure, the docking software, and the chemical library.Screening results: the number of experimentally tested compounds, the hit rate, the hit novelty, the chemical structure of the best hit, the bioactivity, the solved crystal complex structure, the cellular functional study, and the animal experiment.

Note that we only chose the most potent docking hit from each study to simplify the task for potency and structural novelty analysis. We divided the identified docking hits into four groups based on their bioactivities (<1 μM, 1–10 μM, 10–100 μM, and >100 μM). To perform the structural novelty analysis, we retrieved the previously known actives of the specific target (IC_50_, EC_50_, Ki, or Kd ≤ 10 μM) recorded before the corresponding SBVS study from ChEMBL (version 29) [13,14] and Cortellis Drug Discovery Intelligence (CDDI) [17] database. We referred to a target as the widely studied (the number of the previously known active compounds > 100), the less-explored (the number of the previously known actives between 10 and 100), and the least-explored target (the number of the previously known actives < 10). The Tanimoto coefficient (Tc) values were calculated between the docking hit and the previously known actives with Morgan fingerprints using RDKit [18]. If the Tc value of the most similar known active was less than 0.4, the docking hit will be annotated as novel.

## 3. Results and Discussion

### 3.1. Target Profiles

As shown in Figure 1A, about 70% of SBVS campaigns were targeting enzymes, including kinases (73 studies, 57 unique targets), proteases (39 studies, 24 unique targets), phosphatases (20 studies, 16 unique targets), and other enzymes (162 studies, 135 unique targets), followed by membrane receptors (42 studies, 32 unique targets), nuclear receptors (25 studies, 11 unique targets), and transcription factors (12 studies, 10 unique targets). The latest 16 case studies with solid experimental validations of each protein family type were listed in Table 1, and the detailed information of all cases was included in Appendix A. This distribution of protein type classification was tightly related to the availability of structural data, where enzyme structures were relatively easier to determine experimentally. In contrast, membrane receptor structures were more challenging to solve several years ago [19]. Specifically, 238 out of 294 enzymes (81%, including 235 from X-ray diffraction, two from nuclear magnetic resonance (NMR), and one from electron crystallography) studies used experimentally determined structures; and 28 out of 42 (67%) studies of membrane receptors used experimental structures, including 24 from X-ray diffraction, two from NMR, and two from cryogenic electron microscopy (cryo-EM). In general, crystal structures provided a more solid basis for SBVS than modeled structures. However, this may not be the case anymore in the following decades. With the facilitation of the protein structure prediction by AlphaFold 2 [20,21], more modeled structures will undoubtedly be employed in SBVS before the target structures are solved experimentally [22,23]. We also expect molecular docking to play a more critical role in lead discovery due to more available structure information from both experimental and computational sides.

Not surprisingly, almost all the publications were from academic organizations, and only 11 were from private industry. We are aware of the intellectual protection (IP) restrictions that limit publication efforts in the industrial setting, which may significantly underestimate the contribution of SBVS in drug discovery. Nevertheless, more than 50% of SBVS studies were carried out on widely studied targets, of which the number of previously known actives was more than 100 (Figure 1B). Only 22% were carried out on the least-explored targets (the number of previously known actives is fewer than 10). Among these least-explored targets, antimicrobial targets related to reverting bacterial drug resistance account for half, such as the Mg^2+^ transporter in prokaryotic microbes [24], dTDP-deoxy-L-lyxo-4-hexulose reductase in *Mycobacterium tuberculosis* [25], and high persistence A which was the first toxin found to contribute to *Escherichia coli* persistence [26]. Though high-throughput phenotyping screening is the most commonly used strategy for antibacterial hit identification, and modern drug discovery relies more on clear and novel modes of action, which are crucial to meeting the threats created by the emergence of resistance [27]. For example, growing molecular insights in host-pathogen interactions suggest it is promising to develop compounds targeting the nonessential proteins of pathogens because of low selective pressure for developing and maintaining resistance mechanisms [28,29].

We also found several SBVS studies focusing on underexplored human proteins to investigate their biological functions and druggable potential [30,31,32,33]. For example, Hwangseo Part and colleagues reported the first SBVS application to identify novel inhibitors controlling the activity of receptor protein tyrosine phosphatase σ, which may provide a promising therapeutic strategy for neurological diseases [34]. SBVS was also applied to discover potent inhibitors against peroxiredoxin 1 to validate the druggability of peroxiredoxin 1 in leukemia-cell differentiation [35]. Up to October 2020, most human proteins remain to be explored as potential therapeutic targets. Only 10% of human targets have approved drugs or are known to bind small molecules with high potency [36]. Targeting novel proteins should be most significant for docking screening, which provides an efficient yet inexpensive approach to discovering bioactive compounds as chemical probes to study the protein function in cells and organisms [37].

**Table 1 ijms-23-15961-t001:** Representative SBVS applications against different target families.

Target Classification	Target	Docking Software	No. of Hits/Tested Compounds/Library Size	The Activity of the Best Hit	Crystal Structure/Cellular Assay/Animal Test	The Most Potent Compound in ChEMBL
Kinases	HK-2 [38]	GLIDE	5/40/240,000	IC_50_ = 11.3 μM	No/Yes/Yes	IC_50_ = 7.9 nM CHEMBL3806103
	RET^V804M^ [39]	GLIDE	n.a./197/984,000	IC_50_ = 0.02 μM	Yes/Yes/No	IC_50_ = 0.04 nM CHEMBL4067871
Phosphatases	STEP [40]	GLIDE	11/32/240,000	IC_50_ = 9.7 μM	No/Yes/No	IC_50_ = 10 nM CHEMBL3398243
	NAMPT [41]	GLIDE	40/1028/750,000	IC_50_ = 0.22 μM	No/Yes/No	IC_50_ = 0.03 nM CHEMBL566757
Proteases	Caspase 1 [42]	GLIDE	4/50/1 M,	IC_50_ = 0.01 μM	No/Yes/Yes	IC_50_ = 0.01 nM CHEMBL360788
	SAP2 [43]	GOLD	4/50/713,000	IC_50_ = 5.04 μM	No/Yes/Yes	Ki = 12 nM CHEMBL2206678
Other enzymes	IDO1 [44]	FRED	7/50/8.2 M	IC_50_ = 16 nM	No/Yes/Yes	IC_50_ = 0.018 nM CHEMBL432537
	NSDHL [30]	GLIDE	2/495/388,852	IC_50_ = 39.33 μM	Yes/Yes/No	n.a.
Membrane receptors	GPR120 [45]	AutoDock Smina	2/13/350,000	IC_50_ = 23.21 μM	No/Yes/No	EC_50_ = 5 nM CHEMBL3952043
	5-HT_5A_R [46]	DOCK 3.7	5/25/6 M	Ki = 115 nM	No/Yes/Yes	Kd = 0.15 nM CHEMBL323208
Nuclear receptors	PRAPγ [47]	FRED GLIDE	201/2943/70,000	IC_50_ = 24 nM	Yes/Yes/Yes	EC_50_ = 0.018 nM CHEMBL1813011
	AR [48]	GLIDE Autodock Vina GOLD	2/61/210,000	IC_50_ = 10.94 μM	No/Yes/No	Ki = 0.01 nM CHEMBL180681
Transcription factors	TNF-α [49]	ICM-Pro	1/7/6 M	IC_50_ = 0.11 μM	No/Yes/Yes	Ki = 0.03 nM CHEMBL3642435
	p53 [50]	GLIDE	10/244/20,000	IC_50_ = 36.8 μM	No/Yes/No	Kd = 0.12 nM CHEMBL4208820
Others	PqsR [51]	OpenEye docking GLIDE	n.a./500/85,061	IC_50_ = 0.98 μM	Yes/Yes/No	Kd = 210 nM CHEMBL2426244
	ShhN [52]	GLIDE	7/209/200,000	IC_50_ = 1.4 μM	No/Yes/No	IC_50_ = 0.53 nM CHEMBL3091556

n.a. denotes not available. Abbreviations: HK-2: hexokinase-2; STEP: striatal-enriched protein tyrosine phosphatase; NAMPT: nicotinamide phosphoribosyltransferase; SAP2: aspartic protease 2; IDO1: Indoleamine 2,3-Dioxygenase-1; NSDHL: NAD(P)-dependent steroid dehydrogenase-like; GPR120: G-protein coupled receptor; 5-HT_5A_R: 5-Hydroxytryptamine receptor 5A; PRAPγ: Peroxisome proliferator-activated receptor γ; AR: androgen receptor; TNF-α: Tumor Necrosis Factor-α; PqsR: *Pseudomonas* Quinolone Signal LysR-type transcriptional regulator; ShhN: N-terminal product of sonic hedgehog.

### 3.2. SBVS Protocol

We did not find any association between the docking software and the resulting hit potency and hit rate. GLIDE, a commercial docking program, is the most popular docking software used in SBVS, followed by the DOCK series (including DOCK3-6 series), GOLD, and AutoDock series (including AutoDock, AutoDock Vina, and AutoDock Smina) (Figure 2). When comparing the docking software usage in large-scale virtual screening (Table 2, docking library size > 10 million), we found that the DOCK 3 series showed a substantial capacity, especially for ultra-large virtual screening (docking library size > 100 million). Furthermore, the hits discovered in ultra-large virtual screening usually carry a novel scaffold, and the bioactivity is promising. For example, after docking to a library containing 138 million compounds, the identified D4 dopamine receptor hits demonstrated better bioactivities than previously reported binders in experimental evaluation [53]. Thus, it is desirable and encouraging to expand the number of molecules and chemotypes that can be explored in SBVS [54]. However, because the algorithms of sampling and scoring vary from docking software to docking software [5], it is not apparent which docking program is better in general. For example, one recent benchmarking study demonstrated comparable performance between DOCK 3.7 and AutoDock Vina on DUD-E dataset, with Vina outperforming on nuclear receptors and DOCK 3.7 performing better for GPCR and proteases; both approaches have severe limitations on torsion sampling [55]. Thus, docking software and parameters should be carefully evaluated before practical SBVS application.

In addition, a receptor-based pharmacophore model may have great potential in combination with a docking approach to fasten the large-scale docking screening. Site-identification by ligand competitive saturation (SILCS) technology has been successfully applied to identify putative ligand binding sites and generate receptor-based pharmacophore models [56,57]. Further, novel compounds were discovered by combining the SILCS pharmacophore models with virtual screening approaches [57,58,59,60,61].

We want to emphasize that SBVS does not simply prioritize compounds based on the docking scores alone in practice. As many benchmarking results have demonstrated, current docking software is relatively successful in sampling; the scoring power is poor [62,63,64]. Thus, post-docking analysis (including filtering and rescoring) is necessary for selecting compounds for experimental testing. Drug-likeness and pan-assay interference compounds (PAINs) filtering is often used to exclude unwanted compounds. Protein-ligand binding patterns are usually visually inspected, and a clustering approach is always employed to select a diverse set of structural scaffolds [65]. Notably, human intervention may strongly influence result analysis on well-known targets like family A GPCRs [66,67]. The basic amine group forms crucial charge-charge interactions with the binding site aspartic acid, and the top-scoring compounds without amine group would most likely be excluded in the visual inspection stage. Therefore, the superior docking enrichment performance in family A GPCRs may have hidden biases of human knowledge beyond the scoring power. Typically, the docking scores of top-ranked compounds could be re-estimated using other complementary methods, including molecular mechanics with Poisson–Boltzmann surface area or generalized born and surface area solvation, free energy perturbation calculation, quantum mechanics-based calculation, and consensus scoring strategy [68,69]. Most of our reviewed publications used at least one rescoring method to reduce the false positive rate. We believe combining fast but crude molecular docking and a slower but more accurate post-docking rescoring approach is helpful in SBVS applications [8,70,71,72].

**Table 2 ijms-23-15961-t002:** Sixteen large-scale SBVS case studies.

Target	Year of Publication	Docking Software	No. of Hit/Tested Compounds/Library Size	The Activity and Tc of the Best Hit	Crystal Structure/Cellular Assay/Animal Test
AR [73]	2011	GLIDE ICM-Pro 3.6	9/213/10 M	IC_50_ = 38.8 μM Tc = 0.52	Yes/Yes/No
Antithrombin [74]	2012	FlexScreen	1/4/13 M	Kd = 45 nM Tc = 0.20	No/Yes/No
LSD1 [75]	2013	GLIDE ICM GOLD	6/121/13 M	IC_50_ = 19 nM Tc = 0.24	No/Yes/No
Activin [76]	2015	GLIDE SRUFLEX	5/30/10 M	IC_50_ = 3.67 μM Tc (n.a.)	No/Yes/Yes
TRPV55 [77]	2019	GLIDE	2/43/12 M	IC_50_ = 2.91 μM Tc = 0.37	Yes/Yes/No
D4 dopamine receptor [53]	2019	DOCK 3.7	81/549/**138 M**	Ki = 2.3 nM Tc = 0.43	No/No/No
AmpC β-lactamase [53]	2019	DOCK 3.7	5/44/99 M	Ki = 1.3 μM Tc = 0.38	Yes/No/No
KEAP1 [78]	2020	AutoDock series	23/590/**1.3 B**	Kd = 114 nM Tc = 0.26	No/No/No
DNMT3B [79]	2020	GLIDE	5/15/10 M	IC_50_ = 13.5 μM Tc = 0.21	No/No/No
Melatonin receptor [80]	2020	DOCK 3.7	15/38/**150 M**	EC_50_ = 12.89 μM Tc = 0.34	No/Yes/Yes
HSP90 [81]	2021	AutoDock Smina	3/12/13 M	EC_50_ = 0.98 μM Tc = 0.19	No/Yes/No
σ2 receptor [82]	2021	DOCK 3.8	127/484/**490 M**	Ki = 3.9 nM Tc = 0.39	Yes/Yes/Yes
SARS-CoV-2 3CL protease [83]	2022	DOCK 3.7	3/100/**235 M**	IC_50_ = 40 μM Tc = 0.45	Yes/Yes/No
α_2A_AR [84]	2022	DOCK 3.7	17/48/**301 M**	EC_50_ = 52 nM Tc = 0.41	Yes/Yes/Yes
SARS-CoV-2 3CL protease [85]	2022	DOCK 3.7	12/194/**1.2 B**	IC_50_ = 88 μM Tc = 0.32	Yes/Yes/No
SERT [86]	2022	DOCK 3.7	13/36/**200 M**	Ki = 0.92 μM Tc = 0.37	Yes/Yes/Yes

The ultra-large virtual screenings are highlighted in bold. No active compound was identified for Acvidin before 2015. Abbreviations: HSP90: heat shock protein 90; SARS-CoV-2 3CL protease: severe acute respiratory syndrome coronavirus 2 3-chymotrypsin-like protease; LSD1: lysine specific demethylase 1; TRPV5: Transient receptor potential vanilloid 5; KEAP1: Kelch-like ECH-associated protein 1; DNMT3B: Methyltransferase 3 β; α_2A_AR: α2a-adrengeric receptor; SERT: Serotonin transporter.

### 3.3. SBVS Results

To fairly evaluate how valuable SBVS has been in lead discovery, we systematically analyzed the structural novelty and experimental reliability of docking hits individually. The activities of identified hits were better than 100 μM in most studies, among which the best hits in 113 were better than 1 μM (Figure 3A). SBVS, especially for new targets, does not aim to or guarantee the discovery of high-affinity binders, and micromole activity is generally acceptable. In practice, we consider good docking hits to rely more on their structural novelty and derivatization feasibility rather than the binding affinities alone. Nevertheless, the new scaffolds will likely connect new biology and bring unexpected pharmaceutical properties [2].

Encouragingly, the structural similarities between the docking hits and the previously known actives measured by Tc are less than 0.6 in most cases (Figure 3B), indicating that SBVS can be reliably used to identify novel hits. Furthermore, even considering the widely-studied targets with more than 10,000 known actives, such as phosphatidylinositol-4,5-bisphosphate 3-kinase (PIK3CA) [87], σ non-opioid intracellular receptor 1 (SIGMAR1) [88], and neurotrophic receptor tyrosine kinase 1 (NTRK1) [89], the power of exploring much larger chemical space by SBVS is evident.

Docking results should be validated by comprehensive experiments, including but not limited to structural, biochemical, cellular function, and in vivo tests. Complex structure determination is always the gold standard for binding pose evaluation. We analyzed 252 docking screening with decent hit novelty (Tc < 0.4) and previously known actives of more than 10, we found that 17.1% of the studies determined crystal structures, 65.8% carried out cellular experiments, and only 9.5% conducted animal tests, respectively. In great detail, a few successful demonstrations of how SBVS serves as the basis of a drug-discovery process are presented below (Table 3).

**Acetyl-CoA carboxylase (ACC).** Traditionally developed ACC inhibitors lack drug-like properties because the ligand-binding pocket is highly hydrophobic. Thus, SBVS was applied to screen 1.3 million lead-like compounds against a newly discovered allosteric binding site. Among 250 diverse structures, a novel compound ND-022 was selected for the enzymatic tests with IC_50_ of 3.9 μM on hACC1 and IC_50_ of 6.6 μM on hACC2. Based on the crystal complex structure of hACC2 and ND-022 (Figure 4A), a clinic candidate ND-630 was developed with excellent in vitro and in vivo efficacy [90]. ND-630 is undergoing phase II clinical trials in non-alcoholic steatohepatitis (NCT03449446) [91,100].

Targeting allosteric pockets is highly interesting in drug discovery due to the potential benefit of high selectivity. In addition to the ACC case, there are 12 more cases where SBVS was carried out against an allosteric binding site. All identified hits possess satisfactory structural novelty to previously known actives with Tc values around 0.3 (Table 4). The allosteric binding site is structurally distinct from the classic orthosteric binding site, and it is unsurprising that allosteric sites can be used to discover entirely new chemotypes [101].

**Fat mass and obesity-associated protein (FTO)**. FTO is genetically linked to both obesity and diabetes and is an enzyme that demethylates the N^6^-adenosine-modified (m6A) sites in mRNA molecules biochemically [114,115]. Although it is an attractive new biological target, it was unclear whether FTO is a druggable target for which small chemical inhibitors could be developed for treating metabolic disorders. Peng and colleagues addressed this complex problem by performing a docking screening to predict entacapone as an FTO inhibitor from 1323 FDA-approved drugs. Entacapone selectively inhibited the activity of FTO in vitro, significantly reduced body weight, and lowered fasting blood glucose levels in vivo [93]. As the first discovered FTO inhibitor, entacapone was further optimized based on the FTO-entacapone crystal complex structure (Figure 4B) [94,95].

Repurposing of ‘old’ drugs to treat new diseases is gaining increased interest because it involves low-risk and well-characterized compounds with potentially lower overall development costs and shorter development timelines [116]. Entacapone was initially developed to treat a chronic disease (Parkinson’s; in combination with L-DOPA) and has been used safely in clinics for many years. This significant advantage of entacapone, i.e., its safety, would enable repurposing usage. Inspired by this discovery, investigator-initiated trials (IIT) were carried out for the entacapone treatment of obesity (NCT02349243) [96] and gastrointestinal stromal tumors (NCT04006769) [97], individually.

**SARS-CoV-2 3CL protease.** SARS-CoV-2 3CL protease is an essential enzyme in SARS-CoV-2 virus replication and pathogenesis. Shionogi utilized a different strategy than most pharmaceutical companies like Pfizer used to discover oral bioavailable, non-covalent, and non-peptidic 3CL protease inhibitors. By docking screening of an *in-house* compound library, the top-scoring compounds were selected and further prioritized using receptor-based pharmacophore features, including two hydrogen bond acceptor sites and a lipophilic site in the S2 pocket. Finally, one of the hit compounds containing a triazine scaffold was chosen as the lead compound for structure-based optimization (IC_50_ = 8.6 μM), which led to the clinical candidate S-217622 (IC_50_ = 13 nM) [98]. S-217622 was eventually approved in Japan on 22 November 2022. To the best of our knowledge, it may create a history of becoming the first marketed drug that originated from SBVS.

Interestingly, the triazine scaffold perfectly fits into the 3CL pro binding pocket, forming critical hydrogen bond interactions between the carbonyl oxygens of triazine moiety and the main-chain amide groups (Figure 4C). Such a scaffold is rarely presented in commercially available compound libraries, and we located a patent filed by Shionogi, which reported a series of triazine-containing compounds targeting the P2X Purinoceptor 2/3 receptor for treating chronic pain and overactive bladder (WO2010092966) [117]. We hypothesized that the successful discovery of S-217622 strongly relied on the *in-house* compound collection of Shionogi, which included their historically discovered triazine-containing compounds. The size and diversity of the screening library are vital in SBVS practical applications.

**α_2A_AR.** The activation of α_2A_AR has been validated to have pain-relieving effects. However, the existing analgesic therapeutics targeting α_2A_AR also have strong sedation side effects. An ultra-large library containing 20 million fragment-like and 281 million lead-like molecules was screened against the small orthosteric site of α_2A_AR. The notable hit not only behaved 52 nM in functional assay but also showed previously unexplored interactions in crystal complex structure (Figure 4D), and could separate sedation from analgesia in animal experiments. The further optimized lead compound showed better bioactivity and had no sedation side effects [84].

As listed in Table 2, docking hundreds-of-million-level libraries has revealed novel scaffolds for a growing range of targets, against which even nanomolar ligands have been identified. The crucial element of ultra-large virtual screening is preparing a large, diverse library and efficient docking platform. Most molecules in the latest version of ZINC library [118,119] have not been synthesized, but can be made-on-demand. Some other strategies, such as DNA-encoded chemical library technology [120] and generative learning algorithms [121,122], were also developed for generating scaffolds with high diversity, high synthesis availability, and low assay artifacts. In addition to DOCK 3.7, the most preferred docking software in ultra-large virtual screening, an automated and versatile open-source platform, VirtualFlow, was recently built for accelerating hit identification [78]. Thus, it is promising and hopeful to access vast areas of chemical space and identify new chemotypes with high affinity.

## 4. Conclusions

The essence of SBVS is to find structurally novel ligands against previously untapped and well-precedent targets. We have thoroughly analyzed the prospective SBVS case studies within the past 15 years to investigate whether molecular docking hits the bullseye. Based on our findings, most docking screening studies focused on precedented targets, with few cases targeting novel targets, including antimicrobial and human protein targets. However, despite the absence of comprehensive experimental evaluations, most identified hits showed promising structural novelty. We also use four examples to demonstrate genuinely novel chemotypes and solid experimental evaluations that achieved great success in biological research and clinical trials.

Due to the unavailability of studies in the pharmaceutical industry, we only reviewed the case studies published in the literature here. Besides, there must be some omissions in the collection of case studies. However, a detailed picture of SBVS is presented in this review. It is encouraging that docking screening could provide novel ligands against targets of interest and achieve success in practical applications. Furthermore, SBVS will boost hit identification in the immediate future, thanks to the advancements in several areas, including protein structure prediction, allosteric binding site identification, docking library augmentation, sampling and scoring algorithm, and post-processing strategies.

## Figures and Tables

**Figure 1 ijms-23-15961-f001:**
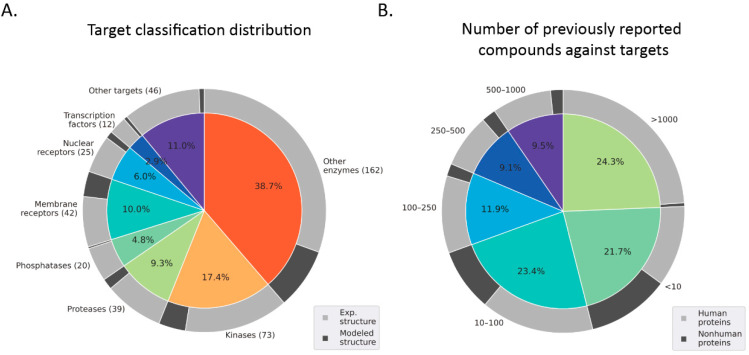
Summary of drug targets employed in SBVS campaigns. (**A**) The inner pie plot represents the distribution of protein type, and the outer pie plot presents the experimentally-determined 3D structures or modeled structures used in docking screening. The number in the bracket is the number of publications. (**B**) The inner pie plot shows the distribution of the number of previously known actives for each target in our assembled dataset, and the outer pie plot represents the human proteins or nonhuman proteins.

**Figure 2 ijms-23-15961-f002:**
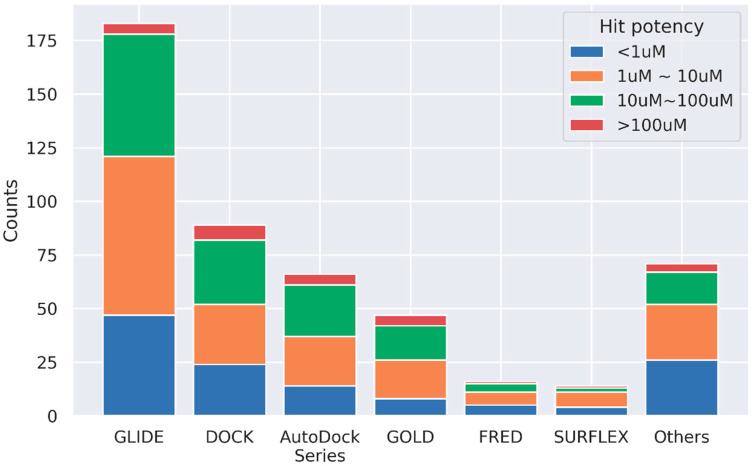
Distribution of docking software usage in our survey. Note that AutoDock Series includes AutoDock, AutoDock Vina, and AutoDock Smina; DOCK includes DOCK 3–6 series.

**Figure 3 ijms-23-15961-f003:**
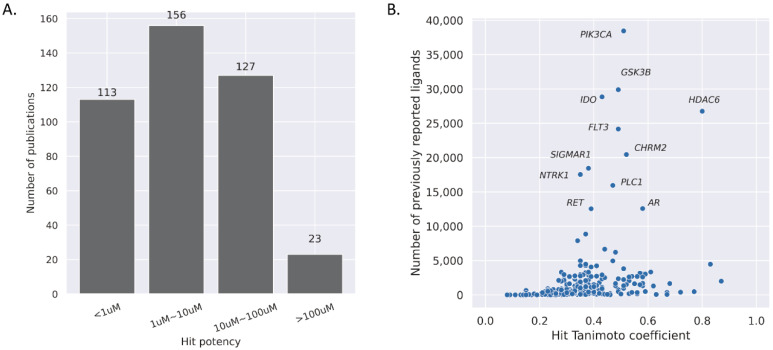
Summary of SBVS results. (**A**) Hit potency distribution. (**B**) Tanimoto coefficient against the number of previously reported compounds. Only the cases with more than ten reported binders were plotted. The highlighted proteins: PIK3CA: Phosphatidylinositol-4,5-Bisphosphate 3-Kinase Catalytic Subunit α, GSK3B: Glycogen Synthase Kinase-3β, IDO: Indoleamine-2,3-Dioxygenase, HDAC6: Histone Deacetylase 6, FLT3: Fms Related Receptor Tyrosine Kinase 3, CHRM2: Cholinergic Receptor Muscarinic 2, SIGMAR1: σ Non-Opioid Intracellular Receptor 1, NTRK1: Neurotrophic Receptor Tyrosine Kinase 1, PLC1: 1-Phosphatidylinositol 4,5-Bisphosphate Phosphodiesterase γ-1, RET: Proto-Oncogene Tyrosine-Protein Kinase Receptor Ret, AR: Androgen Receptor.

**Figure 4 ijms-23-15961-f004:**
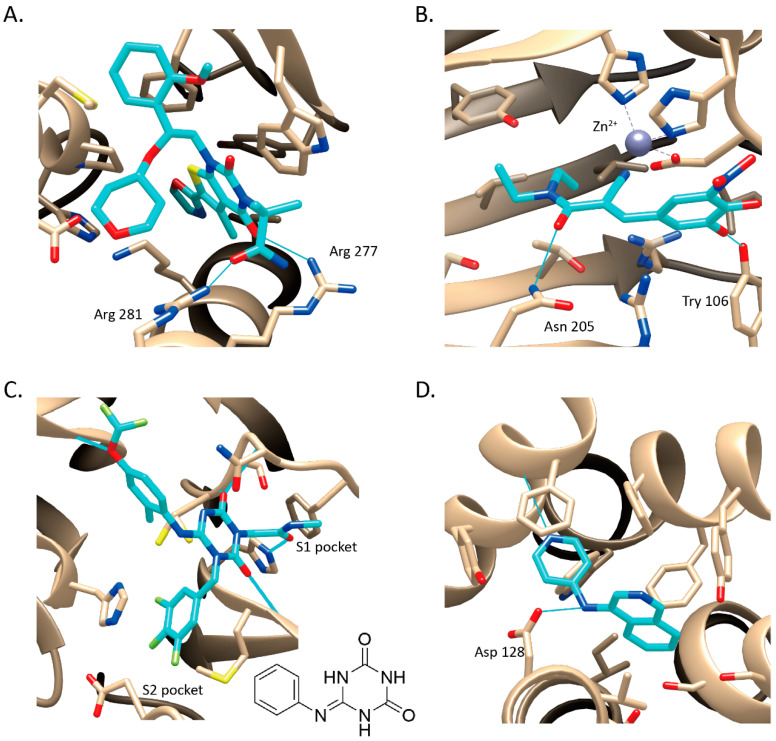
Crystal complex structures were obtained in four successful SBVS studies. (**A**) ACC and ND-646 (PDB ID: 5KKN). ND-646 is the primary amide of the lead compound. Crystallization of the free acid was unsuccessful. (**B**) FTO and entacapone (hit compound, PDB ID: 6AK4). (**C**) SARS-CoV-2 3CL protease and the hit (PDB ID: 7VTH). Triazine scaffold is the critical substructure of the hit and the lead. (**D**) α_2A_AR and ZINC1173879087 (hit compound, PDB ID: 7W6P). All the images are rendered in CHIMERA [123].

**Table 3 ijms-23-15961-t003:** Four representative SBVS studies.

Target	Year of Publication	Docking Software	No. of Hit/Tested/Library Size	Docking Hit	Optimized Compound	Cellular Assay/Animal Test	Clinical Stages
ACC [90]	2016	GLIDE	n.a./250/1.3 M	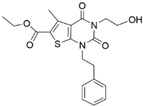 hACC1 IC_50_ = 3.9 μM hACC2 IC_50_ = 6.6 μM Tc = 0.35	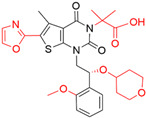 hACC1: IC_50_ = 2.1 nM hACC2: IC_50_ = 6.1 nM PDB ID: 5KKN *	Yes/Yes	Clinical trial: NCT03449446 (Phase 2) [91] and NCT05305547 (Phase 3) [92]
FTO [93]	2019	DOCK 3.5	1/19/1323 (FDA-approved drugs)	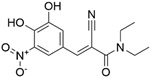 IC_50_ = 3.5 μM Tc = 0.18 PDB ID: 6AK4	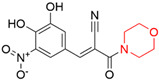 IC_50_ = 0.7 μM PDB ID: 6AEJ	Yes/Yes	Patents: WO2016206573A1 [94] and WO2014082544A1 [95]. Clinical trial: NCT02349243 (IIT) [96] and NCT04006769 (IIT) [97]
SARS-CoV-2 3CL protease [98]	2022	GLIDE	n.a./300/*In-house* library (<1 M)	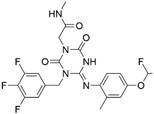 IC_50_ = 8.6 μM Tc = 0.22 PDB ID: 7VTH	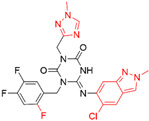 IC_50_ = 0.013 μM PDB ID: 7VU6	Yes/Yes	*JPRN-jRCT2031210350* (Phase 2/3) [99] Approved in Japan on 22 November 2022
α_2A_AR [84]	2022	DOCK 3.7	17/48/301 M (ZINC 15)	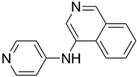 EC_50_ = 52 nM Tc = 0.41 PDB ID: 7W6P	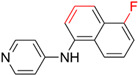 EC_50_ = 4.1 nM PDB ID: 7W7E	Yes/Yes	*n.a.*

* 5KNN is the crystal structure of ACC and ND-646, which is the primary amide of the lead compound.

**Table 4 ijms-23-15961-t004:** Allosteric inhibitors identified in SBVS.

Target	Docking Software	No. of Hits/Tested Compounds/Library Size	The Activity and Tc of the Best Hit
Calpain-1 [102]	GLIDE	3/10/36,503	IC_50_ = 7.5 μM Tc = 0.30
IKKβ [103]	DOCK 6 AutoDock 4	5/133/200,000	IC_50_ = 35 μM Tc = 0.31
CDK2/cyclin A3 [104]	GLIDE	1/15/n.a. (SPECS and Chemdiv libraries)	IC_50_ = 52.1 μM Tc = 0.37
FDPase [105]	GLIDE SURFLEX DOCK5 AutoDock	n.a./n.a./3 M	IC_50_ = 31.6 μM Tc = 0.36
KRAS [106]	GLIDE	9/90/700,000	Kd = 2.74 μM (^GNP^KRAS^WT^) Kd = 3.45 μM (^GNP^KRAS^G12D^) Kd = 8.02 μM (^GNP^KRAS^G13D^) Tc = 0.16
Caspase-6 [107]	AutoDock Vina	5/40/57,700	IC_50_ = 30.2 μM Tc = 0.25
M_2_ mAChR [108]	DOCK 3.6	3/13/4.6 M	EC_50_ = 1.1 μM Tc = 0.27
Protein kinase PDK1 [109]	DOCK 3.6	2/3/6300	Kd = 39.1 μM Tc = 0.23
PHGDH [110]	GLIDE	4/98/n.a. (SPECS library)	IC_50_ = 0.56 μM Tc = 0.15
PGDH [111]	DOCK 6 AutoDock Vina	2/170/n.a. (SPECS library)	IC_50_ = 34.8 μM
SIRT6 [112]	GLIDE	2/20/5 M	EC_50_ = 173.8 μM Tc = 0.06
GPR68 [113]	DOCK 3.6	4/17/3.1 M	EC_50_ = 0.11 μM

No active compound was reported previously against PGDH and GPR68. Only one active compound, an acyclic peptide, was collected against SIRT6 before 2018, two actives were collected against PHGDH before 2016. Abbreviations: SIRT6: sirtuin 6; GPR68: G protein-coupled receptor 68; M_2_ mAChR: muscarinic acetylcholine receptor M2; KRAS: kirsten rat sarcoma virus; CDK2: cyclin-dependent kinase 2; IKKβ: I kappa B kinase β; FDPase: fructose 1,6-bisphosphatase; PHGDH: *Homo sapiens* phosphoglycerate dehydrogenase; PGDH: D-3-phosphoglycerate dehydrogenase (*Escherichia coli*).

## Data Availability

Detailed information of 419 prospective SBVS case studies is openly available at: https://docs.google.com/spreadsheets/d/1x_hO3CrgD6i8sHZMVupO0z4_DQs_0Rab/edit#gid=2073190557.

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
