# Peer review of "A Comprehensive Survey of Prospective Structure-Based Virtual Screening for Early Drug Discovery in the Past Fifteen Years"

_ijms, 2022, doi:10.3390/ijms232415961_

Round 1

Reviewer 1 Report

Zhu et all present an overview of structure-based screening studies in

which the hit compounds have been experimentally validated. The

manuscript is well written and generally focused on standard docking

approaches. The information content of the manuscript will be of

interest to scientists working in the field of CADD, making the

overview suitable for IJMS. Some issues that the authors should

address are listed below.

A paragraph on problems associated with non-experimentally validated

screening studies would be a good addition to the paper. For example,

novel compounds presented in such papers are now known to the art such

that the may no longer be considered as intellectual

property. Accordingly, it is higly unlikely that such compounds will

be pursued both other laboratories even though they may be quite

interesting. Essentially, such compounds have been removed as

candidates for development into drugs.

The SILCS technology should be included in the survey. It represents a

novel approach that has successfully been applied to multiple targets

yielding experimentally validated hits as seen in the following

studies. Notable is the ability to identify agonists vs. antagonists

as well as novel allosteric modulators of B2AR.

It may be worth noting the inherent bios in docking studies targeting

receptors of biogenic amines in which the selection of compounds with

basic nitroges favors successful identification of active

compounds. Note that this includes the opioids as well as

Reviewer 2 Report

The review is well written and can be helpful to novel scientist in Structure Based Drug Discovery. The authors have covered most of the known advantages and disadvantages of Docking Simulations studies. 1) Docking simulations are ideal to retrieve new scaffolds. 2) Most publications presented gave new hit compounds with Tc similarity less than 0.6, 3)Table 4 summurize some of the success stories of such drug design campaigns.

One note to the authors: On table 1, I would suggest to include the activity of the most potent compound found in CHEMBL for comparison.
